# Constructing Territories of Deterritorialization–Reterritorialization in Clarice Lispector Novels

**Fátima Velez de Castro** 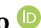

Department of Geography and Tourism, University of Coimbra, 3004-531 Coimbra, Portugal; velezcastro@fl.uc.pt

**Abstract:** The importance of geography and literature, as producers of knowledge for society, is undeniable. Both areas are structural pillars for the explanation of contemporary territorial phenomena. In this article, we intend to reflect on the importance of literature for understanding migration, focusing on the deterritorialization–reterritorialization process. Through geographic lenses, we will perform a content analysis of several fictional works by Clarice Lispector, who was herself a migrant, in several moments of her life. We can consider that this writer's contribution made a deep contribution to (re)think conceptual and theoretical frames in the geography of migration.

**Keywords:** literature; geography; migrations; deterritorialization; reterritorialization; Clarice Lispector





## 1. Introduction

The importance of the humanities and social sciences, specifically geography and literature, as producers of knowledge for society, is indisputable. Both areas are structural pillars for the understanding and explanation of contemporary territorial phenomena. Rossetto (2014, p. 513) refers to the resurgence of the Literary Geographies as an "( . . . ) emergent arena of geohumanities is strongly connected with the extraordinary growth of new geographical technologies", like cartography or even areas like tourism (geographic literary routes). Matozzi (2019) refers to the importance of the recovery of past cultural memories and the continuities of the present, in a logic of transdisciplinarity that necessarily implies epistemological and methodological readings and dialogues that are differentiated, but at the same time are supportive and complementary. Thus, it will be possible to explore, in a more complete way, the dynamics of contemporary migrations, looking for empirical clues in literary works with a biographical and/or fictional character.

The role of literature in society is decisive to create communities that are more aware of themselves, as well as of these "Others" that are part of their surroundings. In this respect, Claval (2006) understands "the book" as the instrument that generates a geography of collective memories. As a cultural manifestation (literature as a domain and the book as an instrument), they act in an active and pragmatic way in the way of understanding and acting in the world. As a structuring pillar of collective culture, Mitchell (2007) states that it is a very important sphere of human life, which can interfere in other spheres, such as politics, economy, or society. André (2020) refers to literature and the book as instruments generating the possibility of dialogue, contestation, questioning, response, observation and understanding "the other". Additionally, Souza (2011) highlights the role of books, reading and text comprehension, as a desirable approach in geographic education, by using mechanisms such as perception, memory, and observation of social and spatial practices.

This article aims to achieve the objective to reflect on the importance of literature for the understanding of geographical phenomena—the deterritorialization and reterritorialization process. To achieve this purpose, it will be used as a literary analysis methodology that favors geographical knowledge. Thus, it will be focused on a geo-literary path of a particular author—Clarice Lispector—in the Geography of Migrations, considering her importance and national and international projection, as a renowned writer.

With a profoundly reflective and existentialist literary production, she intensely scrutinizes the psychological intimacy of the characters she creates, who represent individuals with apparently banal daily lives, in momentary tears that come together to form a narrative whole. This constructive–structural characteristic of his work, as a narrative strategy, is very important for the reader and for the researcher, since it seems to allow certain aspects of the stories, places, and characters to be highlighted in a particular way. The narration "in instalments" is often short, intense, and incisive, giving the necessary clues to understand the phenomenon addressed more clearly, cleared of any obscure or less necessary contours. For the study in question, which focuses on the understanding of migratory experiences of reterritorialization, such a structure facilitates, above all, the understanding of the typification of the characters, as well as their performance.

The theme of migrations—internal (north-eastern) and external (international)—appears transversally, associated both to the paths or mobility projects, as well as to the characteristics of those involved. Being herself a migrant in three moments of her life, she seems to reflect, consciously or unconsciously, her experiences, not so much in the first moment, when she accompanies her family in the displacement between Europe and Brazil, but rather in the second, when she migrates, with her father and sisters, from Recife to Rio de Janeiro. Internal migrations, namely the north-eastern flows towards the metropolises in the south of the country, gain special prominence, becoming the central topic of the novella "The hour of the star"[1] (1977), and the suburban-center mobility is addressed in the novel "The besieged city"[2] (1949). The third moment, related to her experience as a diplomat's wife in Europe and the United States, is reflected in the observations made by the characters, when alluding to aspects of the landscape that they can experience, or that they recall when reflecting on their own experiences. On the one hand, his work reflects his experience as a migrant. On the other hand, the author considers herself as an observer. In this respect, it assumes the role of autochthonous, assuming a purely Brazilian territorial identity. This idea is in line with what Clarice Lispector assumes in several moments of her life: Brazil as "her" country and the Portuguese language as "her" language. Therefore, the author will always start from an observation influenced by this geographical territoriality. Velez de Castro (2021) deepens the study of this author's works, highlighting her in the context of other Brazilian authors who address the theme of the deterritorialization–reterritorialization process in their novels.

In terms of organization, the article was structured in an initial part, where it is made as an approach of the migration theoretical and conceptual framework that supports the reading of Clarice Lispector' s novels. Then, it presented the methodology used in the analysis of the novels, followed by the analysis and discussion of the literary work of Clarice Lispector in the light of the Geography of Migrations.

Next, we will understand the conceptual dimension inherent to the works under analysis, which allowed us to establish an analytical parallel between Clarice Lispector's literature and the theme of deterritorialization–reterritorialization.

## 2. Approach of the Migration Theoretical and Conceptual Framework That Supports the Research

While the literature provided the object(s) of study—the narrative(s)—Geography oversaw integrating the territorial vision on the subject under study—migrations—namely the central concepts "territorialization-deterritorialization-reterritorialization". Deleuze and Guattari (1980), followed by the studies of Haesbaert (2003, 2004) presents territorialization in a logic of topophilia, i.e., based on the affective, emotional and identity relationship that individuals establish with territories. This spatial dimension is part of the individual's identity, acting the everyday space of presence as an element of balance, of ontological security. However, when individuals move and arrive at the migratory destination place, there is a loss of the territory where the migrant was inserted, involved, integrated. This process, called deterritorialization, can be considered as the loss of character and the absence of topophilic relations with the place of migratory destination, the inability of the

migrant to interpret, understand and integrate in its daily dynamics, can lead to the feeling of loss of territory, the disintegration of the initial identity.

However, the deterritorialization process is not ad aeternum. The precariousness of territoriality ends up being a finite process, as power relations are eventually established with the surrounding context, in a logic of identification of subjects and territorial dimensions (Fernandes et al. 2016). Moreover, there is no space without time, which means that, during the stay, one ends up establishing a relationship with the territory of arrival, in a logic of reterritorialization. Tuan (2018) draws attention to the fact that the relationship between mobility and the sense of place is not always easy and immediate. It is necessary to (re)establish a routine, which involves contemplating, interpreting, and knowing everyday places, ranging from the house, the neighbourhood, the city, and the region. At the heart of these, there are micro-territories that need to be incorporated into one's identity, as is the case of the home, workplace, leisure spaces, school, market, etc. (Figure 1).

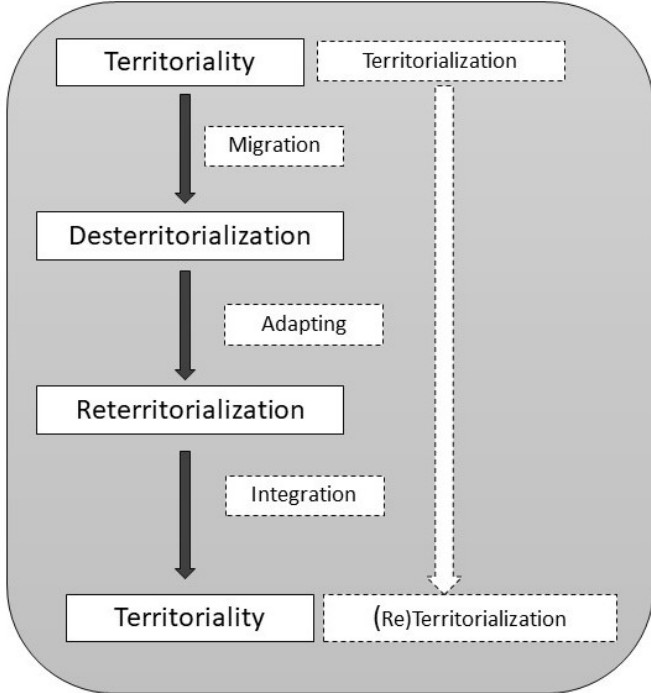

**Figure 1.** Central concepts of the study.

Around these concepts, there are others that helped in the understanding of the processes of deterritorialization–reterritorialization. This is the case for the migratory project, understood as the plan drawn up by the migrant, which presupposes the achievement of a set of goals, which may vary according to aspirations, but which may go through diverse areas such as employment, housing, education, social condition, economic level, among other aspects. It should also be noted that the migratory project is also built based on the territorial image, which is constituted based on a set of expectations about the territory of the migration destination. These derive from a construction that is based on information originating from third parties, with migratory experience, as well as from various sources, such as the media or social networks.

It is also important to mention the attraction–repulsion theory, since it was this that was at the heart of the conception of the current migratory theoretical framework. Conceived by Ernest Ravenstein at the end of the 19th century and revised by Everett Lee in the 1960s, it advocated that those migrations occur because individuals experience negative situations—repulsive factors—in the place of origin, perceiving positive situations—attractive factors—in the migratory destination place. The labour dimension may assume a very important role in what concerns migratory motivations.

These concepts are linked to the assumptions of deterritorialization and reterritorialization, since the conception of the idea of the place of origin and the approximation (or not to reality) are factors that determine the interaction and the level of preparation to interpret the place of arrival.

Therefore, the concept of mobility capital is important for the study. This refers to the capabilities and skills acquired by individuals, which arise from a multiplicity of mobility experiences, which can derive from simple leisure or business trips to dynamic migratory movements, which include adjacent displacements or temporary returns to the place of origin (for example, on holiday). This manifestation will translate into a dynamic of multi-territoriality, i.e., considering the perspective of Haesbaert (2004), it is an expression that derives from multiple territorial experiences, in a logic of synthesis. In this sense, the migrant manages to synthesize in himself several spatial dimensions, combining territories of experience, such as that of origin and migratory destination, as well as multiple travel experiences with various purposes (work, leisure, etc.).

In the works under analysis, it is possible to reflect on fictionalised experiences of characters, which manifest in their experience processes of deterritorialization and reterritorialization.

## 3. Methodology in the Analysis of the Novels

The novels of Clarice Lispector that were analyzed in this paper refer to three domains. The Novel: "The besieged city" (see note 2) (1949), "An apprenticeship or the book of pleasures"[3] (1969), "Água viva"[4] (1973), "Breath of life"[5] (1978); the Novel: "The hour of the star" (see note 1) (1977); the Short Story: "The via crucis of the body"[6] (1974). The choice was made because these works present characters and plots related to the migratory dynamics in Brazil, revealing individual characteristics of migrants, as well as related social, geographical, and cultural paths.

A qualitative methodological dimension guided this work, which is assumed to be based on "content analysis". First, we must take up the idea of Stake (2016, p. 58), who assumes that "The function of research is not necessarily to map and conquer the world, but to sophisticate its contemplation". This principle is the basis of an investigation committed to the principle of reading, analysis, and reflection, which underpinned this work. It means that relevance was given to the selection and discussion of contents, with the aim of clarifying and bringing to public knowledge actions expressed in the work of five Brazilian authors, which could contribute to the study of the processes of migratory reterritorialization in their country.

For the analytical context in question, attention was paid to the advantages of qualitative research, allowing for the "causal explanation (...) of objects, events or situations not necessarily considered representative", being considered advantageous because of the fact that "relationships can be established in many and varied situations" (Heindrich 2016, p. 24). In this sense, the content analysis seemed to be the most appropriate methodology for the work proposed in the literary works, for allowing a plastic and flexible approach to the meaning of the discourse and the context of the stories, privileging the view of geography on the facts presented.

Content analysis can be defined as a research technique that aims to make replicable and valid inferences obtained through the reading of texts of varied content (fiction, news, etc.). It means that the researcher should consider the inherent and underlying content of the text, but also the meanings that derive from the interpretation performed (Krippendorff 2004). Shurmer-Smith (2002) draws attention to the importance of symbolic meanings, namely their decoding, as well as to the analysis of the narrative, i.e., on the point of view of the person who produces the discourse.

Hsleh and Shannon (2005) refer to research using content analysis that may start from pre-defined categories or these same categories may be conceived from the reading and interpretation of the text in question. In the case of the study presented here, and although it was based on previous knowledge around the geography of migrations, it was based on

a "clean" basis, only limited by the theme proposed for study—the construction of literary territories from migratory experiences of reterritorialization.

In this sense, the position of Guerra (2014) was considered, in which categories were identified based on concepts and theoretical assumptions in migrations, and these were linked to explanatory–associative ideas evident in the text, which helped in the embodiment and revelation of the dynamics in question. In this way, the possibility of comparison was assumed to be at the core of the work itself, and also externally, between the various works analyzed (Bardin 2019).

By way of conclusion to this chapter, Bryman (2016) warns that content analysis is often accused of being "unscientific", since nowadays quantitative procedures are privileged. However, in this work, a qualitative, analytical, and reflective perspective is taken, as it allows for a more holistic and transdisciplinary approach to the works under study.

## 4. Analysis and Discussion of the Literary Work of Clarice Lispector in the Light of the Geography of Migrations[7]

### 4.1. "The Besieged City"[8] (1949)

Let us begin the analysis with this work and the presentation of its protagonist. Lucrécia is a young girl who lives in a suburb—S. Geraldo—but who aspires to move to the big city, aiming to improve her standard of living, especially in terms of material goods. This social and economic ascension ends up being materialized through the marriage with Mateus, which provides displacement and residence in the city, as well as a comfortable economic situation.

The work focuses, in a comparative logic, on the suburban and metropolitan daily dynamics, which is being carried out by the experiences and reflections of the protagonist, so one can understand the existence of two distinct parts in the story: one of living in the suburb (where she says she cannot escape reality); another of presence in the city (where she perceives the surroundings based on the distorted territorial image, which she built based on expectations). This is not a story in which an international migratory movement is presented, but rather a phenomenon of rural exodus, carried out within the country itself, on a regional scale. However, and considering the characteristics of this type of flow and its similarities with international migratory dynamics, it seemed pertinent to approach "The besieged city" (1949) as the beginning of a study that focuses on the theme of deterritorialization/reterritorialization of individuals in the face of places of mobility.

In the first part of the work, Clarice Lispetor deals with Lucrécia's life in the suburbs. The story begins at a party in S. Geraldo, where the population behaves effusively and noisily: "The girl could not stand this free laughter which was a way for the outsider to despise the poor festivity of S. Geraldo. (...) Where would the center of the suburb be?" (Lispector 2009, p. 8). There is a permanence, which is accompanied by the construction desire for the city of a territorial image of what she conceives as urban space. In a dream, she seems to return to the origin of places, of civilization, of things, for "(...) she had gone back until she was dressed in long skirts and smoothing hair bands on her forehead (...) [She was] Greek in a city not yet erected (...) And her destiny as a Greek was as unconscious as now in S. Geraldo. What was left from so far away? what was left of Greece? the insistence (...)" (Ob.Cit., p. 65). In other words, the dream is the manifestation of the desire for the city since the origin of Western civilization, since always. There is an attraction of the city as a destination, and a repulsion of the suburb, understood to some extent not as an origin, but as a point of passage.

She dates Perseus and Filipe, without effective manifestation of affection for both, since they themselves belong and represent the suburbia, for which she feels repulsion. Perseus is calm and gentle, but "(...) dresses like a farmer. And the girl was already in need, in her iron streets, of armed force" (Ob.Cit., p. 43). While this character studies, manifesting knowledge of natural phenomena far beyond what happens in the suburbs, Lucrécia does not seem to truly believe that he can or even aspire to leave there. Already with Lieutenant Filipe, both try to distance themselves from the connection to the suburb and act as if they

were superior to that territory. In a violent argument, Filipe humiliates Lucrécia by making a geographical reference to her origin, reminding her that she is part of that place, and he is not, because he is the outsider. When he tries to give her a kiss and she violently refuses he says "-Uneducated is what you are! (...) And it's my fault for hanging around with people like that, these must be the manners of that filthy suburb of yours! he said with pleasure, insulting her right in her own city. (Ob.Cit., p. 42).

In the second part of the work, Clarice Lispetor presents Lucrécia in the city, who achieves the desired social ascension through marriage and the consequent migration: "With pondering she looked from one side to the other, calculating and measuring this new city she had bought. Like the ambitious young women of S. Geraldo, waiting for the wedding day to free her from the suburbs (...)" (Ob.Cit., p. 85); "She also wanted to waste no time in looking at the new city—yes, the real metropolis—which would be the prize of the stranger—every man seems to promise a bigger city to a woman" (p. 86). However, what she aspired to in her plans for the city did not come to pass, but rather became embodied in perfectly alienating circumstances, the result of the loss of the original territory—deterritorialization: "(...) it is destiny, she was content to be subjugated. (...); "From the incomprehension of the market street, she had gone on to public incomprehension" (Ob.Cit., p. 92). He then begins to compare what he found repulsive in his native land and concludes that the familiarity of the suburbs and belonging to a known community would not be repeated in the city: "(...) he calculated on the new landscape, comparing it with the one in S. Geraldo" (Ob.Cit., p. 93); "On the crowded pavements no one would look at her, whose pink dress would still be of interest in S. Geraldo. (...) Once out of the suburb, her kind of beauty disappeared, her importance diminished. (...) Lucrécia Neves had begun by being anonymous" (Ob.Cit., p. 86).

The decision to return is grounded in reflections based on the negative experience of alienation provided by the urban space. This idea is evident in the time of the decision, when "One-night Lucrezia cried a little (...). Then she said in anger: I'm leaving here. In the hope that at least in S. Geraldo "street would be street, church, church, and even horses would have rattles" (Ob.Cit., p. 97). But the return implies a new adaptation—reterritorialization. S. Geraldo has also changed, going himself from suburb to city: "Taking advantage of her [Lucrécia's] absence, S. Geraldo had advanced in some direction, and she no longer recognized things. When she called them, they no longer answered—they were used to being called by other names (...) She let herself be guided by her husband, as if she were the foreigner in S. Geraldo" (Ob.Cit., p. 97).

Ironically, Perseus migrates, and ends up working in the city as a doctor in the central hospital. We may compare Lucrezia's attitude of loss and resignation, as opposed to her ability to be resilient, i.e., to adapt and transform herself in the face of the evolution of territories and societies, by assuming the logic in which her migratory act is transverted: "(...) and even if one gets lost—to get lost is also a path" (Ob.Cit., p. 138).

*4.2. "The Hour of the Star"[9] (1977)*

The last work, written by Clarice Lispector, is a literary reference document that is set around the North-eastern migrations. The author, who tells the story of a poor orphan from the Northeast—Macabéa—seems to identify, in part, with the character, since she herself, her sisters, and her father, were also part of this internal Brazilian migratory flux. However, when writing this novel, she integrates her own youthful migratory experience, with the vision of an adult and cosmopolitan woman. Its origin, its original territoriality, even if in a discreet way, is never forgotten. This is visible through the typology of the characters she creates, which largely reveal what she herself was or could have been.

She also confronts the readers with this unequivocal probability when she writes: "How is it that I already know everything that will follow and what I still don't know, since I never lived it? It is that in a street in Rio de Janeiro I caught in the air briefly the feeling of doom on the face of a girl from the Northeast. Not to mention that as a boy I grew up in the Northeast. I also know things because I am living. Those who live know, even

without knowing that they know. That is why the masters know more than they imagine and are pretending to be sound" (Lispector 2002, p. 14). The truth is that not everyone likes to assume certain emigratory pasts, a fact that occurred in Clarice Lispector's life, but of which she redeems herself, as she refers that "I seem to know in the smallest details this North easterner, for if I live with her" (Ob.Cit., p. 24). In other words, she projects her life, creating an alter-ego in which she hides to make known her own account of being a migrant.

The story revolves around the young Macabéa, a 19-year-old North easterner who flees the life of poverty of her place of origin to try her luck in the big city. The narrator emphasizes the initial living conditions of the character to justify the displacement, by stating that "(...) to talk about the girl I must not shave for days and acquire dark circles under my eyes because I sleep little, only napping from sheer exhaustion (...). Besides dressing in torn old clothes. All this to put me at the level of the north-eastern" (Ob.Cit., p. 22).

The narrative preparation is preceded by an account of the early years of Macabéa, an orphan raised by a bizarre aunt, which greatly limited the aspirations as well as the worldview developed by the protagonist. She says that "She was born entirely rickety, a heritage of the Sertão (...). Then—it is unknown why—they had come to Rio, the unbelievable Rio de Janeiro, her aunt had got her a job, finally died and she, now alone, lived in a room shared with four other girls clerking at the American Stores" (Ob.Cit., pp. 31, 33).

This type of migration fits in the attraction–repulsion theory of Ernest Ravenstein, who argues that individuals tend to feel negative factors in the place of origin, and positive in the migratory destination place, so they tend to move. In this case, a scenario of initial poverty is presented, which seems to justify the genesis of migration, combined with the construction of a distorted territorial image of the place of migration destination. The concept of territorial image corresponds to a set of expectations and ideas built around the place of migration destination, because of information from various sources (news, emigrant friends, etc.), which give rise to the construction of a distorted image about that same territory.

Other than questioning the true nature of the migratory flow, that is, if in fact migrants manage to improve their standard of living in the city, the narrator reinforces this idea, through the alienation of the young girl in the non-place of the urban space, by stating that "The North easterner was lost in the crowd" (Ob.Cit., p. 44). The concept of non-place refers to undifferentiated territories, aesthetically dubious, in which the constituent elements seem to live around an alienating and undifferentiated daily life. Apparently, there is no development of a topophilic relationship, i.e., no real affective ties are established between the living territory and its users.

As a migrant, Macabéa's daily life is limited to various spaces of use: the shared room where she lives with other young women like her, who despise her; the office, where she works oppressed by her boss; the pier, the only place where she has her scarce leisure time during the week, which is the destination for her Sunday walks. Also, the public space, where she lives with her boyfriend: "They sat on what is free: a bench in a public square. And settled there, nothing distinguished them from the rest of nothing" (Ob.Cit., p. 52). That is, to a large extent the young woman lives in spaces of repulsion (bedroom and office) or else in spaces of social exclusion (wharf and park bench), considered inferior by the natives.

Macabéa's character is in stark contrast to that of her boyfriend—Olímpico—who is also a migrant in the big city: "He came from the Sertão da Paraíba [region] and had a resistance that came from his passion for the wild land cracked by the drought" (Ob.Cit., p. 62). While different, they recognize themselves as belonging to the same diaspora, when it is observed that: "The boy and she looked at each other through the rain and recognised each other as two north easterners, beasts of the same species that sniff each other" (Ob.Cit., p. 47).

This contrast comes from the differences established by the migratory project of each one of the characters. Macabéa only survives in an immediate daily logic, not seeking an end to her existence. The narrator clarifies that: "(...) Macabéa, in general did not worry about her own future: having a future was a luxury" (Ob.Cit., p. 63). Olímpico, on the other hand, aspires to an obvious social ascent, hoping to enter the world of politics and become rich, soon, by stating: "I am very intelligent, I am still going to be a state representative" (Ob.Cit., p. 50).

It should also be added that Macabéa is also distinguished from Olímpico by the way the process of deterritorialization–reterritorialization occurs. To "deterritorialize" means to lose the original territorial references in a migration, so that the individual tends to rebuild his identity and his daily life, reterritorializing himself in the migratory destination. Now Olímpico, starting a relationship with a girl from Rio de Janeiro and aspiring to a political life, as the ruler of social life in the city to which he migrated, shows signs of resilience, that is, of adaptation and identity reconstruction. With Macabéa, however, the same does not seem to happen, since the character, in addition to not being able to leave the spaces of exclusion, also fails to create relationships with the indigenous community. Loneliness is a patent characteristic, as can be seen in some of the descriptions made by the narrator: "(...) this girl who slept in a denim combination (...). To fall asleep in the frigid winter nights she curled up in herself, receiving and giving herself the meagre warmth. (...) she was incompetent. Incompetent for life. She lacked the knack of fitting in (...)" (Ob.Cit., pp. 25–26). "(...) One of her roommates didn't know how to tell her that she smelled like a dying man. (...) Nobody looked at her on the street, she was cold coffee" (Ob.Cit., p. 30).

Regarding the work situation, the narrator gives details about the nature of the work, eminently manual, undifferentiated, and poorly paid: "Olímpico de Jesus worked as a labourer in a metalwork (...). Macabéa was happy with his social position because she was also proud to be a typist, although she earned less than the minimum wage. (...) The work [of Olímpico] consisted of picking up metal bars (...). He had never wondered why he put the bar down" (Ob.Cit., p. 49).

Throughout the work there is only one brief reference to the relationship between social networks and employment. This refers to the fact that migrants, as diaspora, can manifest actions that demonstrate mutual help and solidarity within the group (constituents) and outside (potential migrants), namely through the facilitation of displacement through migratory channels, or of establishment at destination (employment, housing, etc.). Such a phenomenon is manifested when "His only kindness to Macabéa was to tell her that he would get her a job in the metalworks when she was laid off" (Ob.Cit., p. 63).

Returning to the initial question, around the characterization of Macabéa, Clarice Lispector confronts the reader with a question: is it worth migrating? Macabéa leaves a context of poverty and loneliness to live in another context of poverty and loneliness. This is noticeable in some moments of the work, when the narrator relates that: "Sometimes before going to sleep I felt hungry and would get a bit hallucinated thinking about cow's thigh. The remedy then was to chew paper well chewed and swallow". (Ob.Cit., p. 35); "(...) the luxury she allowed herself was to take a cold sip of coffee before going to bed" (Ob.Cit., p. 37); "She had never had dinner or lunch in a restaurant. It was standing right at the corner bar" (Ob.Cit., p. 43). The young girl, who was so poor, only ate hot dogs, as it was the cheapest meal she could consume. The visit to the doctor and the initial diagnosis of tuberculosis corroborates the deterioration of the young woman's state of health, the result of the unsanitary housing conditions and the harshness of the working conditions, which together with a poverty unable to confer the minimum conditions of nutrition, had voted her to the disease.

At the end there seems to be a slight hope, with the trip to the fortune teller and the reading of a bright future. The possibility of marriage to a gringo and the reference to the possession of material goods, almost places her at the level of Olimpico's aspirations, leaving the young girl "(...) pregnant with future" (Ob.Cit., p. 85). However, the immediate death in the street, the result of being run over, differentiates the nature of migration

and the consummation of the migratory project, leaving in the air the possible answer to the writer's question with another question. Migrating can represent a risk: is it worth being lived?

*4.3. "An Apprenticeship or the Book of Pleasures"[10] (1969)*

Lori, a young primary school teacher from a rich family, migrates to the big city. It is there that she meets Ulysses, a philosophy professor, who becomes her mentor, and with whom she reflects in depth on what her life is and what meaning lies within it.

In this work, images of two moments in Clarice Lispector's life recur: one in which she carries out an internal migration (with her sisters and father, from Recife to Rio de Janeiro); another in which she carries out an external migration (when she accompanies her diplomat husband, who works in several European countries and in the United States).

In the first moment of the analysis, let us pay attention to the reference to the city-countryside migration (rural exodus), where Lóri assumes, with confidence and transparency, the connection to her place of origin: "(...) she was now a big city woman, but the danger is that there was also a strong agrarian heritage coming from afar in her blood. (...) no, I do not want to be me only, by having a self, I want is the extreme connection between me and the friable and fragrant earth. (...) she wanted the land of her ancestors" (Lispector 1999, p. 38).

In this framework, the motivations for migration are made known, different from the context of poverty expressed at other times in the work of Clarice Lispetor, for example in "The hour of the star"[11] (1977). In this case there is a desire for freedom, to break ties with the oppression of the family sphere, which resulted in expectations about marriage. It is mentioned that "(...) she had moved alone from Campos to Rio (...).—Why did you come to Rio? Are there no primary schools in Campos?—I didn't want... I didn't want to get married; I wanted a certain kind of freedom that would not be possible there without scandal, starting with my family, there everything is known (...)" (Ob.Cit., p. 42). In this case, migration is assumed as a positive and liberating process. The character states that: "It was a funny and good experience to go from the big family rooms, in Campos, to the tiny flat that all of it would fit inside one of the smaller rooms. I had the impression of having returned to my true proportions. And to freedom, of course" (Ob.Cit., p. 82). The apparent spatial loss is replaced by confinement to a small flat, whereby Lóri assumes that freedom is superior to spatial and material constraints. The character identifies with the city, making her believe that the process of reterritorialization has met her aspirations and her true nature. It is mentioned that "- But in your travels it is impossible that you have never been among orange trees, sun and flowers with bees. (...)—No, she said sombrely. Those things are not for me. I'm a big city woman" (Ob.Cit., p. 43).

In any case, the doubt remains whether it was an incomplete reterritorialization, since he assumes: "He was from Campos, a land without sea, and he had never got into the habit of going to the beach, which was so close to his flat" (Ob.Cit., p. 65). This idea is not seen as something negative, but rather as representative of a process of territorial and identity reconfiguration, in which elements of the various territories through which he passes, which manifests itself in a multi-territorial paradigm.

In the second moment of the analysis, one can find several references to the European space. They are geographical flashes that come to Lóri's memory, fruit of past trips, which make her associate her states of mind with places she knows. For example, she recalls "( . . . ) The silence of Switzerland, for instance. She remembered with nostalgia the time when her father was rich, and they travelled several months a year. (...) The night in the mountains is so vast. So unpopulated. The Spanish night has the perfume and the harsh echo of tap dancing, the Italian night has the warm sea even if it is absent. But the Bern night has silence (...) If only there were wind. Wind is anger, anger is life. But on the nights, I spent in Bern, there was no wind and every leaf was embedded in the branch of the still trees" (Ob.Cit., pp. 30, 31).

However, Lóri also has moments when she feels the negative experience of deterritorialization, the loss of her territory and identity. This was identical to what happened to Clarice, who disliked her first migratory experience as a diplomat's wife in Europe. One can digress around the traumatic experience that was the initial migration she made with her sisters and parents, from the territory that is now Ukraine to Brazil. The Lispector family are fleeing from the terror of the progroms, that is, from the scenario of violence perpetrated by the Russian army, and although the author is only two months old when she undertakes the migration, this fact will mark her for the rest of her life. Perhaps therefore she projects it onto Lóri, who relates the French experience as something negative. She says that "She had spoken of Paris, but not of the land called Paris. She spoke of how the winter there was full of darkness in the twilight and how bad it snowed (...)—In that corner, she had said to Ulysses in her always gentle voice, I felt lost, saved from some shipwreck, and thrown onto a dark, cold, deserted beach. Paris, suddenly, that strange land, had given her the most unusual pain—that of her real doom" (Ob.Cit., p. 38).

This work has a happy ending, in which both characters evolve and find meaning in their lives, bringing them together. In this case, Clarice makes the point that migrating is worthwhile. It can be worthwhile.

### 4.4. "Água Viva"[12] (1973) and "Breath of Life"[13] (1978)

These two works, unlike the ones previously analysed, no longer focus on the theme of migration. Even so, Clarice Lispector allows her experience as a migrant to stand out, by narrating brief passages related to the description of places and the respective sensations revisited.

In the case of "Água viva" (1973), the author questions and wanders around her human condition and existence. After reflecting on the future and the possibility of one day experiencing the delicacy of life, she talks about the pain of flowers as a metaphor to give order to her reasoning. He says that "Tulips are only tulips in Holland. A single tulip simply isn't. It needs an open field to be. (...) The wheat flower is biblical. In the nativity scenes in Spain, it is not separated from the branches of wheat (...) Geranium is a window-bed flower. It is found in São Paulo, in the district of Grajaú and in Switzerland" (Lispector 2012a, pp. 47–48).

It is as if each place had its own identity and that same identity could only genuinely subsist in that same place, even if similar territories may exist. This is the case of the geranium, which can be found in Brazil as well as in Switzerland. Apart from the possible ubiquity of the species, one can infer around the effect of migration, with respect to the material and immaterial diffusion of identities, which end up giving a multi-territorial dimension to people and places.

In "Breath of life" (1979) Clarice creates a dialogue between herself and Angela about the value of existence, in an introspective logic. It is in the synthesis about the core of her personality, that the author assumes her vulnerability, fragility and suffering in the face of the deterritorialization process, by assuming that "If I uproot myself, I am left with my roots exposed to the wind and rain. Friable." (Lispector 2012b, p. 23). In fact, the process of deterritorialization can be painful and traumatic, and this happened to Clarice Lispector when she lived in Europe as a diplomat's wife. This idea is strongly projected in the character she creates for dialogue. She states that: "Angela—I am individual like a passport. (...) Should I be proud to belong to the world or should I disregard myself for?" (Ob.Cit., p. 37); "An Angela Pralini? the unhappy one, the one who has suffered a lot. I am like a foreigner anywhere in the world. I am of the never." (Ob.Cit., p. 50); "Angela is afraid to travel for fear of losing herself on a journey." (Ob.Cit., p. 61).

She fears that the multi-territorial experience, rather than enriching her identity, will make her lose what she considers to be her "me", and that she will lose her originality. From this perspective, Clarice Lispector underestimates the power of the mobility capital associated with migrations, i.e., the skills and competences gained by the individual from her multiple experiences of travelling and staying in diverse places.

*4.5. "The Via Crucis of the Body"*[14] *(1974)*

It in this novel that one can find references to Portuguese emigration through the description of the characters. In the short story "The man who appeared", he introduces Manuel, owner of a bar, whose first name and sector of activity were quite associated with the Lusitanian community. He states that "It was Saturday afternoon, around six o'clock. Almost seven. I went downstairs and bought some coke and cigarettes. I crossed the street and went to the Portuguese Manuel's bar". (Lispector 2006, p. 163).

Other than the bar owner, another Portuguese type appears—the housemaid/caretaker—in the story "Mauá square": "Celsinho had adopted a four-year-old girl (...) She lacked nothing: she had everything of the good and the best. And a Portuguese nanny". (Ob.Cit., p. 185).

In addition to the reference to the labor dimension, Clarice Lispector introduces the issue of physical characteristics and the interrelationship between immigrant communities. She refers to "Joaquim [who] was fat and short, of Italian descent. A Portuguese neighbor had given him the name Joaquim. His name was Joaquim Fioriti. Fioriti? a flower had nothing of his own". (Ob.Cit., p. 186).

In this line of ideas, she writes the short story "Better than burning", which is dedicated to Portuguese emigration, from the female point of view. Referring to the protagonist, he states that "She was tall, strong, hairy. Mother Clara had a dark beard and deep, black eyes (...) Mother Clara was the daughter of Portuguese and secretly shaved her hairy legs" (Ob.Cit., p. 191). It happens that this nun questions her vocation, so she decides to leave the monastic life and return to secular daily life. She continues her way, changing her life project which ends up being that of marriage and maternity. This idea can be seen in the following excerpts: "Her black hair grew thick. (...) She went to the botequim to buy a bottle of Caxambu water. The owner was a handsome Portuguese man who was charmed by Clara's discreet manner. He did not want her to pay (...). The Portuguese man, by name António, plucked up courage and invited her to go to the movies with him. (...) "They got married in the church and in the civil ceremony. (...) They went to spend their honeymoon in Lisbon. (...) She returned pregnant, satisfied, happy. They had four children, all men, all hairy" (Ob.Cit., pp. 192–93). In this case, physical characteristics such as robustness and fertility stand out, as well as the masculinity represented by the hairy vastness.

Between the Portuguese migrant and the migrant from the northeast there seems to be a physical opposition that contrasts, respectively, robustness versus fragility, financial stability versus poverty, and satisfaction versus sadness. Notwithstanding the danger of generalizations and stereotyping, these are still valid clues for the analysis of the dynamics of the different migratory communities and how they transformed and were transformed by the reterritorialization territories in Brazil. The author makes no reference to aspects that might lead us to identify the deterritorialization process. Instead, it focuses on a later moment, where the reterritorialization itself has already taken place. Therefore, what is presented to the reader are characters with new territorialities, different from the original ones. It means that the identity verified in the departure territory is not the same. These characters go through the process of deterritorialization–reterritorialization and change. This results in new territorialities, where aspects of the original identity merge with the new cultural and social paths of the arrival country.

## 5. Conclusions

The role of the humanities and social sciences, specifically geography and literature, as producers of knowledge for society, is very important and necessary. Both areas are structural pillars for the understanding and explanation of contemporary territorial phenomena, i.e., migratory systems. This proves that geography, as a science, can and should resort to other areas of knowledge, such as literature, to better understand geographical phenomena. Without ever losing sight of the spatial dimension, which determines its scientific and pedagogical identity, it is desirable that has a dialogue with the fictional text, making it the alibi of a powerful geography.

This article aimed to achieve the objective of reflecting on the importance of literature for the understanding of geographical phenomena—the deterritorialization and reterritorialization process. On the one hand, his work reflects his experience as a migrant. On the other hand, the author considers herself as an observer. In this perspective, it assumes the role of autochthonous, assuming a purely Brazilian territorial identity.

From Clarice Lispector's literary novels, geography can reflect on the dimension of migrations and migrants' territoriality, especially on the deterritorialization–reterritorialization process. Of Ukrainian origin, she arrives in Brazil as a baby, accompanying her family. The stories she writes portray the author's point of view, as a migrant in her early life, in her adolescence and youth (from Recife to Rio de Janeiro), and as an adult, accompanying her diplomat husband who works in Europe and North America. The psychological, social, and territorial complexity of the characters, the aspirations, and territorial constraints, allow a unique understanding about the geographical aspect of migration in Brazil, but that can be generalized to other spatial contexts. This idea is in line with what Clarice Lispector assumes in several moments of her life: Brazil as "her" country and the Portuguese language as "her" language. Therefore, the author will always start from an observation influenced by this geographical territoriality.

The author indicates important aspects that reveal the process of deterritorialization–reterritorialization of the characters, indicating the difficulties of adapting to the destination territories. However, it also reveals modes of integration, focusing on the construction of new territorial identities.

**Funding:** This research received no external funding.

**Institutional Review Board Statement:** Not applicable.

**Informed Consent Statement:** Not applicable.

**Data Availability Statement:** Available online: https://www.researchgate.net/publication/351461406_A_construcao_de_territorios_literarios_a_partir_de_experiencias_migratorias_de_reterritorializacao_O_encontro_entre_a_Geografia_e_a_Literatura_na_obra_de_autorases_brasileirasos (accessed on 5 December 2022).

**Conflicts of Interest:** The author declares no conflict of interest.

## Notes

[1] "A hora da estrela", in the original.
[2] "A cidade sitiada", in the original.
[3] "Uma aprendizagem ou o livro dos prazeres", in the original.
[4] That means, in English, "living water".
[5] "Um sopro de vida", in the original.
[6] "A via crucis do corpo", in the original.
[7] The novels analysed in this article were in the orginal language of publication—Portuguese. To facilitate the reading, all the cited sentence were translated to English language.
[8] "A cidade sitiada", in the original. The quotes in this subchapter, and the corresponding pages indicated, are taken from the following reference: Lispector (2009). A Cidade Sitiada. Relógio D'Água.
[9] "A hora da estrela", in the original. The quotes in this subchapter, and the corresponding pages indicated, are taken from the following reference: Lispector (2002). A Hora da Estrela. Relógio D'Água.
[10] "Uma aprendizagem ou o livro dos prazeres", in the original. The quotes in this subchapter, and the corresponding pages indicated, are taken from the following reference: Lispector (1999). Uma Aprendizagem ou O Livro dos Prazeres. Planeta De Agostini.
[11] See note 1 above.
[12] That means, in English, "living water". The quotes in this subchapter, and the corresponding pages indicated, are taken from the following reference: Lispector (2012a). Água-Viva. Relógio D'Água.
[13] "Um sopro de vida", in the original. The quotes in this subchapter, and the corresponding pages indicated, are taken from the following reference: Lispector (2012b). Um sopro de vida (Pulsações). Relógio D'Água.

[14] The quotes in this subchapter, and the corresponding pages indicated, are taken from the following reference: Lispector (2006). Contos de Clarice Lispector [A Legião Estrangeira; Felicidade Clandestina; A Via Crucis do Corpo; Onde Estiveste de Noite]. Relógio D'Água.

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
