# Peer review of "Constructing Territories of Deterritorialization–Reterritorialization in Clarice Lispector Novels"

_socsci, doi:10.3390/socsci11120575_

Round 1
Reviewer 1 Report
The text focuses on the problems of territoriality of the migrant population, based on Clarice Lispector literature. The author builds a geographical and social conceptual framework, where he describes and explains various aspects of migration theories in a coherent and well-articulated way.
The paper presents a conceptual scheme related to the original territory/deterritorialization/reterritorialization that accompanies the entire analysis.
The objectives are well defined and the author clearly justifies, explains and supports the option for a qualitative approach and methodology, assuming a content analysis in this investigation.
In the areas of ​​Geography and Sociology it is more common to use a quantitative approach. Nevertheless, it is always important to follow other paths, equally valid and necessary to the full understanding of the phenomena.
The author presents a style that oscillates between literary analysis and the lens of geographical knowledge, achieving a research approach. It is
The paper is written in clear English that completes a very interesting and unusual investigation.
Author Response
The Reviewer 1 doesn`t suggest changes in the article.
In any case, it will be made the changes suggested by the other Reviewers (2, 3, 4).

Reviewer 2 Report
I consider that the text presents a good discussion about the relationship between Geography and Literature and that it is also quite original when analyzing Clarice Lispector's works from the perspective of territorialization, de-territorialization and re-territorialization processes. I suggest reviewing the mention about various objectives in the text. It seems to me that there is clearly a single objective which, in turn, was very well achieved.Author Response
Reviewer 2 - "I suggest reviewing the mention about various objectives in the text. Is seems to me that there is clearly a single objective which, in turn, was very well achieved."
The main objetive of the article was highlighted, as suggested by the reviewer.

Reviewer 3 Report
The paper would be improved greatly by integrating the ideas of territorialization and re-territorialization with the analysis of Lispectors' novels, essays, and short stories. What is it that we learn about migration through her literature? How does she depict the migrants? Do her literary works offer contrasting visions of internal migration in comparison to immigration/emigration? Thinking of these questions would help you to integrate the first part with the second part of the paper. As it reads now, the article seems like it is two distinct works: one on the theory of migration and territorialization-reterritorialization, and another on the works of Lispector. I would urge you to avoid generalizations about geography, literature, and literatures of migration. Instead, it might be fruitful to think more specifically of how Lispector crafts geographies of Brazil and its migration through her literary works. How does she characterize a migrant like Macabéa in comparison to herself as a cosmopolitan traveler/migrant (wife of a diplomat)? In the analysis, it would be useful to move beyond what happens to how it happens, to have more than just an analysis of plot. I would also encourage you to look at secondary scholarship about Clarice Lispector. There are existing essays about Clarice, her works, and migration that could be helpful. You might also want to consult discussions of her work as an example of Jewish literature (with Jewish literature being tied to diasporas and, therefore, deterritorializaiton and reterritorialization). Many of the works you cited have been translated into English. Use the published translations, rather than your own, when possible. If the translations are your own, indicate that in the footnote.
Author Response
REVIEWER 3:
1) "The paper would be improved greatly by integrating the ideas of territorialization and reterritorialization (...) [in] short stories". - This was done an helped to improve chapter 4.3.
2) "If the translations are your own, indicate that in the footnote". - The footnote is presented in page 5.
3) "It might be fruitful to think more specifically of how Lispectors crafts geographies of Brazil and its migration throught her literary works". - This idea is developed in page 2.
4) "(...) a migrant like Macabea in comparison tho herself as a cosmopolitan traveler (...)" - this idea is developed in page 6.

Reviewer 4 Report
The manuscript intends to "reflect on the importance of literature for the understanding migration", focusing on the "deterritorialization-reterritorialization process, and performing "a content analysis of several fictional works by Clarice Lispector".
The topic is relevant and contributes to the understanding of Brazilian literature worldwide. However, some considerations on the submitted manuscript must be made:
1) Title: It could be shorter, with one of both phrases from the original as follows:
Literature, Geographies and Migrations [in Clarice Lispector novels]
Constructing territories of deterritorialization-reterritorialization in Clarice Lispector novels
2) Language: the article has to be proofread as language mistakes can be found.
Examples:
Abstract: "focusing [on] the deterritorialization- reterritorialization process"
Also the repetition of the same words/terms should be avoided in the same sentence.
Example: "several fictional works by Clarice Lispector, that was itself a migrant, in several moments of her life"
"this writer contribution made a deep contribution to (re)think"
3) Citation:
Regarding citation, the author hasn't followed a pattern when refering to the quoted writers.
Sometimes, mentions name + surname, as well as his/her position (e.g. "the geographer Paul Claval"); sometimes refers to the author only by his/her surname ( e.g. "André (2020)"). It would be interesting to keep a standard considering that it's an academic text.
4) Objectives:
According to the manuscript,
"This article aims to achieve several objectives, namely: to reflect on the importance of Literature for the understanding of geographical phenomena – deterritorialization and re-territorialization process; to use literary analysis methodology that favours geographical knowledge; to relate the geo-literary path of a particular author - Clarice Lispector - in Geography of Migrations, considering her importance and national and international pro-jection, as a renowned writer."
Due to its broad scope, this goals would be reachable if we were talking about a dissertation/thesis instead of a 13-page paper. Unfortunately that doesn't happen in the manuscript, where some of the goals are mentioned, but not developed with the expected depth, method and criteria.
5) Introduction:
In the first paragraphs, the author discusses the connections between geography and literature, mentioning the "dynamics of contemporary migrations", as well as the "role of Literature in society". Then it passes briefly to the article's goals, and, finally to writer Clarice Lispector's life and production. Only at the very last paragraph of the introduction, the reader gets some information on the methodological approach for the article, but with no details or accuracy.
From a holistic analysis, one observes that the introduction could have been better planned and ordered as it doesn't create a logical structure for a scientific article.
6) Section 2:
Firstly, it brings to light the "central concepts 'territorialization-deterritorialization-reterritorialization' ". Authors such as Fernandes, Trigal & Sposito (2016), Tuan (2008) and Haesbaert (2004) are refered to at this point.
Surprisingly, the original concepts by Delleuze and Guattari are not mentioned of added to the theoretical approach which opens the body of the article.
One strongly recommends the study of "A Thousand Plateaus", where the concepts of territory, (de/re)territorialisation, as well as movement and becoming are consistently worked with.
Also the title for section 2 - "Approach of the migration theoretical and conceptual framework that supports the reading of Clarice Lispector’ s novels" - doesn't correspond to the content as Clarice Lispector's work is not analysed there.
7) Section 3:
Section 3 brings the ambitious project of working with 6 books by Lispector, including 2 different literary genres: the novel and shorty story:
"The novels of Clarice Lispector that were analyzed in this paper refer to three domains. The Novel: "The besieged city”(1949), "An apprenticeship or the book of pleasures “(1969), "Água viva"(1973), " Breath of life"(1978); the Novel: "The hour of the star” (1977); the Short Story: "The via crucis of the body"(1974)."
However, some studies are quite superficial when compared to the complexity of Lispector's literary legacy - mainly the sub-sections listed below:
4.4. “Água viva” (1973) and “Breath of life” (1978)
4.5. “The via crucis of the body” (1974)
This can be considered a downside of the manuscript: too wide, too shallow.
7) Conclusion:
Comprised of 3 paragraphs which try to recapture the connections between literature and geography in Lispector's works; the final assumptions, however, seem to be quite general and superficial when compared to the objected raised at the beginning of the manuscript.
One example follows:
"The psychological, social, and territorial complexity of the characters, the aspirations, and territorial constraints, allow a unique understanding about the geographical aspect of migration in Brazil, but that can be generalized to other spatial contexts."
In this regard, the objectives mentioned in the manuscript (as shown above) have only been partially met. As a suggestion, one recommends a re-analysis of the manuscript goals in order to reduce the scope and produce a deeper analysis within the proposed topic. The method and the conclusions should also be revised in detail.
Author Response
REVIEWER 4
1) "Title" - The title became "Constructin territories of deterritorialization-reterritorialization in Clarice Lispector novels."
2) "Language" - It was made a general language revision of the text.
3) "Citation" - The citation was strandardized (p.1)
4) "Objectives" - as it was suggest also by reviewer 2, it was assumed just one objetive, that was reached in the article.
5) "Introdution" - As it was suggested, it was presented the structure of the article (page 2).
6) "Section 2" - It was introduced the reference to Delleuze and Guattari in the text (p.2) and to the respective work "A Thousands of Plateaus"(p.12). Also the title of section 2 was ligthly changed, to reflect the content (p.2).
7) "Section 3" - It was introduced the reference of a deep study, were the author of the article developed the analyses of Clarice Lispector`novels (p. 2 and p.12).
8) "Conclusion" - The structure of this part of the article changed and some new ideas were introduced (p.12)

Round 2
Reviewer 4 Report
Some changes have been made in the manuscript in accordance with the reviewers' comments and suggestions. However, as pointed out before ( Topic 7 of the previous review), Section 3 may lack the required depth of analysis, which turns to be the text downside.
The acceptance will depend on the other reviewers' viewpoint.